# Three Polymers from the Sea: Unique Structures, Directional Modifications, and Medical Applications

**DOI:** 10.3390/polym13152482

**Published:** 2021-07-28

**Authors:** Lei Wang, Wenjun Li, Song Qin

**Affiliations:** Key Laboratory of Biology and Bioresource Utilization, Yantai Institute of Coastal Zone Research, Chinese Academy of Sciences, Yantai 264003, China; Bio_wangl@163.com (L.W.); wjli@yic.ac.cn (W.L.)

**Keywords:** alginate, chitosan, collagen, biomaterials, tissue engineering

## Abstract

With the increase of wounds and body damage, the clinical demand for antibacterial, hemostatic, and repairable biomaterials is increasing. Various types of biomedical materials have become research hotspots. Of these, and among materials derived from marine organisms, the research and application of alginate, chitosan, and collagen are the most common. Chitosan is mainly used as a hemostatic material in clinical applications, but due to problems such as the poor mechanical strength of a single component, the general antibacterial ability, and fast degradation speed research into the extraction process and modification mainly focuses on the improvement of the above-mentioned ability. Similarly, the research and modification of sodium alginate, used as a material for hemostasis and the repair of wounds, is mainly focused on the improvement of cell adhesion, hydrophilicity, degradation speed, mechanical properties, etc.; therefore, there are fewer marine biological collagen products. The research mainly focuses on immunogenicity removal and mechanical performance improvement. This article summarizes the source, molecular structure, and characteristics of alginate, chitosan, and collagen from marine organisms; and introduces the biological safety, clinical efficacy, and mechanism of action of these materials, as well as their extraction processes and material properties. Their modification and other issues are also discussed, and their potential clinical applications are examined.

## 1. Introduction

Wound hemorrhage, tissue defects, and body damage caused by various physical, chemical, and human factors are common clinical phenomena. In the process of treatment, many medical wound repair materials are needed as an aid. Bioremediation materials, especially natural biomedical materials, are a class of safe and effective materials. They are widely used, due to their good biocompatibility and cell adhesion. The most representative type is animal-derived collagen, which has been developed into many varieties of soft tissue and hard tissue repair materials; however, the effects of zoonotic diseases, religious ethics, etc., restrict the large-scale use of animal-derived biomedical materials [1]. With the continuous deepening of human development of the ocean, marine biological materials without pollution, comorbidities, and religious issues have attracted widespread attention [2].

Chitosan is a derivative of chitin after deacetylation. With good biocompatibility, bioactivity, biosafety, biodegradability, and hypoallergenicity, it is called a multipotent biomaterial. It has excellent physical and chemical properties, such as high specific surface, porosity, tensile strength, and electrical conductivity, it is prepared into different products and dosage forms (such as films, fibers, sponges, powders, powders, gels, solutions, etc.) that are widely used in clinical processes [3]. Alginic acid is a kind of natural polysaccharide, which is widely found in brown algae. Due to the different amounts of guluronic acid (G) and mannuronic acid (M) in the molecule, the composition and relative molecular mass are diverse, making it salts and mixtures show functional diversity, such as high hygroscopicity, easy removal, high air permeability, gel blocking, biodegradability, compatibility, and metal ion adsorption [4,5], which are widely used in the field of wound hemostasis and repair. Collagen, as a kind of medical material widely used in the field of tissue engineering, has achieved good application effects for soft tissue and hard tissue repair. Marine animal collagen has a similar amino acid sequence structure to terrestrial mammal collagen, and has similar biocompatibility, safety, and degradation properties to terrestrial animal collagen. It is an excellent source of collagen for medical products [6,7].

However, as these three marine-derived biomedical materials are widely used clinically, their shortcomings and deficiencies are gradually being exposed: for example, single-component chitosan has a low antibacterial ability, fast degradation speed, poor water solubility, etc. [8]; single alginic acid has poor cell adhesion, too strong hydrophilicity, a slow degradation rate, poor mechanical properties, etc. [5,9]; and pure collagen products have poor mechanical properties and rapid degradation in the body, plus the lack of uniform production quality standards, inspection standards, etc. The development of marine-derived biomedical materials is restricted [10,11]. How to obtain high-quality, high-performance clinical products using the aspects of source control, molecular structure adjustment, extraction process improvement, and molecular modification, and expand their clinical application range, has become a hot research topic. This article summarizes the raw material sources, structure and function relationships, main effects and mechanisms, shortcomings, and modification methods of the three marine biomedical materials of chitosan, alginate, and collagen, and their clinical applications and application prospects are discussed.

## 2. Sources, Unique Structures, and Functional Groups 

### 2.1. Alginate

Alginate mainly exists in the cell wall and matrix of brown algal cells, and plays a mechanical supporting role for cells [12]. It is mainly extracted from kelp and exists in the form of calcium alginate, magnesium alginate, potassium alginate, and sodium alginate, etc. [13]. Different types of brown algae have different molecular weights [14] and contents of alginate, and the differences increase with changes of temperature, marine environment, and season. 

Alginate is a kind of polyanionic natural hydrophilic polysaccharide, which is mainly composed of d-mannuronic acid (M) and l-guluronic acid (G) [15] (Figure 1, Table 1). It has powerful water imbibition, is insoluble in water and non-polar solvents, and contains free carboxyl groups with active properties. Many hydroxyl groups create strong hydrophilicity. These structural characteristics make alginate a good moisturizer. Combined with metal ions, which are monovalent or polyvalent, it is converted into alginate, which has good gelatinizing and hemostatic properties [16,17]. The immunogenicity of alginate, which can induce higher levels of cytokines for wound healing, is affected by the M-unit amount in the alginate.

### 2.2. Chitosan

Chitosan, the product of the *N*-deacetylation group of chitins, is mainly extracted from the shells of shrimp, crabs, and other crustaceans (Figure 1, Table 1). It is the only natural alkaline polysaccharide discovered so far and belongs to the group of linear sugars [25]. Its features include being white-translucent, solid with a pearl luster, its relative molecular mass ranging from thousands to millions, and being insoluble in water and alkaline solutions, but soluble in dilute acid solutions, such as formic acid and acetic acid. Chitosan is a natural non-toxic biopolymer that has three functional groups, which are the amino/acetyl amino on C-2, the hydroxyl on C-3, and the hydroxyl on C-6. The amino group on chitosan is the main cause of the unique physical and chemical properties of chitosan. The protonation of the amino group makes chitosan carry a positive charge and induces the agglutination reaction of bacteria or red blood cells by electrostatic attraction, thus achieving bacteriostatic effects and hemostasis. The gluco-1, 4-glycoside bond in the chitosan molecule is easily degraded into *N*-acetyl-glucosamine by lysozyme, chitosanase, and other enzymes secreted by cells, and then induces and promotes the processes of tissue self-repair. In addition, the degree of deacetylation and molecular weight of chitosan have a greater impact on physical and chemical properties and biological properties, such as solubility, hydrophilicity, and cellular response (Figure 1, Table 1).

### 2.3. Collagen

Marine collagen is a macromolecule with a triple helix structure that is mainly obtained from invertebrates and vertebrates such as sponges [26,27,28], fish [29,30], squid [31] and echinoderms [32,33], and extracted from the skin, bones, scales, muscles, and other tissues [34].

Studies have shown that the amino acid composition of marine fish collagen is like that of terrestrial mammals. Glycine accounts for about 30% of the total amino acid content, while proline hydroxyl accounts for 35% to 48%. The contents of methionine, isoleucine, and tyrosine are slightly lower than those of land mammals [35]. In addition, the presence of many diamino-dicarboxylic groups makes collagen extremely hydrophilic and hemostatic (Figure 1, Table 1).

## 3. Mechanisms for Main Functions

### 3.1. Alginates

#### 3.1.1. Hemostatic Mechanisms

Alginate types of hemostatic materials are mainly calcium alginate or other alginates bound with divalent ions, which can achieve hemostasis through absorption, ion exchange, etc. Studies have shown that in the coagulation mechanisms of calcium alginate: (1) The –COOH on the alginate molecular chain can react with the NaCl in the blood, causing the ionization balance to be broken and the coagulation factor to be activated. Sodium alginate absorbs a large amount of water in the blood, making the blood viscosity increase and the flow rate slow down. Moreover, the generated gel can block the end of the capillary, having the effect of ballistic hemostasis [36]. (2) After the alginate fiber absorbs the blood, the calcium ions and sodium ions are exchanged to rapidly form the gel. The calcium ions are released into the blood as coagulation factor IV, which together with other coagulation factors activates prothrombin into thrombin and participates in the endogenous coagulation process [37]. (3) Calcium alginate contains plant agglutinin, which aggregates red blood cells, and then causes red blood cells to transform from discs into leaves, exposing phosphatidylserine on their surface and promoting local prothrombin conversion into thrombin [38] (Figure 2).

In addition to these clotting mechanisms, researchers added zinc ions to alginate fibers, which showed effects of enhance clotting and platelet activation.

#### 3.1.2. Wound Regeneration Function of Alginates

Calcium alginate fiber has high hygroscopicity and a good gel-forming performance, so it can quickly absorb exudate and blood from the wound and form a low-viscosity gel to cover the wound surface, effectively keeping the wound wet; providing a suitable environment for cell migration and regeneration of blood vessels; and, finally, accelerating wound healing [39]. Many studies have shown that calcium ions released by calcium alginate fibers can stimulate the growth of fibroblasts and facilitate cell migration to the wound site, before attempting accelerating wound healing and repair. Additionally, researchers have pointed out that there are plentiful calcium ions in the cells of keratinocytes and sebaceous glands around the wound surface. It can be inferred that calcium alginate fibers may participate in the whole healing process through the release of calcium ions. The efficacy of calcium alginate fiber in inducing and promoting wound regeneration has been proved for acute and chronic wounds [40,41] and post-skin grafting care. At the same time, alginic acid gel can also repair various soft and hard tissue wounds by carrying various cells, factors, etc. (Figure 3).

#### 3.1.3. Antibacterial Function of Alginate

After the alginate is implanted into the wound, it reacts with sodium ions in the blood to quickly form a dense hydrogel, and an imbibition reaction occurs. Alginate bloating, on the one hand, causes bacteria at the wound to follow the liquid into the sodium alginate hydrogel, thus restricting its free movement. On the other hand, it results in less liquid at the wound site and limits the range and activity of bacteria in order to inhibit bacteria from forming [42] (Figure 4).

#### 3.1.4. Function as Carriers

The gelatinizing property of alginate gives it unique advantages in both drug embedded and sustained release. Due to its anion carboxyl group in its molecular structure, it is often made into microspheres with cationic amino chitosan, and the ratio between these two components determines the stability of the microsphere structure and the time span of drug sustained release [43]. The strong hydrophilicity of alginate can weaken the stability of microspheres and affect the loading amount and sustained-release effects of drugs, so its application in sustained-release drugs is limited without modifications [44,45]. For example, it is necessary to prepare microspheres with liposomes for enhanced hydrophobicity and to graft with dodecyl glycidyl ether (DGE) to make amphoteric carriers.

### 3.2. Chitosan

#### 3.2.1. Hemostatic Mechanisms

The hemostatic effect of chitosan has been widely applied clinically, especially for superficial wounds. Its coagulation mechanisms mainly include: (1) Chitosan is a kind of alkali polysaccharide with positive charge. After implantation into the wound, it can adsorb the red blood cells with negative charge, to gather and form a thrombus and seal the wound surface and induce hemostasis [46]. (2) Chitosan can replace damaged tissue at the wound surface because of its good adhesion, promote platelet activation, and thus shorten the hemostasis time [47] (Figure 5).

#### 3.2.2. Wound Regeneration

Chitosan is a kind of aminoglycan composed of glucosamines connected by a β-1, 4-glycoside bond. Its structure is like that of glucosamine in the human body, with good biocompatibility, non-toxicity, and non-immunogenicity. The mechanisms of the wound repair function of chitosan (Figure 6) are due to: (1) Chitosan can be degraded to *N*-acetyl-β-d-glucosamine by lysozyme and other enzymes secreted by cells. The degraded products can be absorbed by tissues and organs, promote the orderly deposition at the wound site, and stimulate the synthesis of hyaluronic acid, thus stimulating the proliferation of fibroblasts and the formation of blood vessels, resulting in an acceleration of wound healing [48]. (2) Chitosan can activate macrophages, promote the secretion of various factors, accelerate the phagocytosis of fragments, prevent other abnormal growth activities, and thus improving the overall repair of wounds. At the same time, chitosan can inhibit the secretion of collagen I, promote the synthesis of collagen III, reduce the contraction rate of the wound, and thus reduce the formation of scar tissue [49]. (3) Chitosan can be combined with other polymers to give better mechanical properties and spatial structures for cell migration, crawling, proliferation, and growth. Chitosan can be loaded with cells, factors, and drugs to accelerate wound healing [50].

#### 3.2.3. Antibacterial Function

The antibacterial mechanism of chitosan is closely related to its amino group. There are many factors influencing its antibacterial effects, including proton concentration, deacetylation degree, molecular weight, and pH value, etc. [47,51]. The antibacterial mechanism of chitosan (Figure 7) is mainly speculated to have the following forms: (1) The free amino protonation gives chitosan a positive charge, while the cell wall of most bacteria has a negative charge. Thus, bacteria are attracted to chitosan and unable to move normally, and finally flocculation occurs [52]. (2) Low molecular weight chitosan with –NH^3+^ can pass through bacterial cell walls and membranes in an acidic environment, interfering with the normal replication and transcription of DNA, and thus affecting the replication and reproduction of bacteria [53] (Figure 2). (3) Free amino groups on the surface of chitosan can chelate with metal ions and trace elements in an environment, and chelate with cofactors of various enzymes in bacteria. Thus, the uptake of trace elements and enzyme activities in vivo and t bacteria growth are affected [54].

#### 3.2.4. Function as Carriers

Chitosan can be prepared into microspheres with a suitable particle size and excellent drug loading and sustained release ability through emulsification, crosslinking, evaporation in a solvent, in-liquid drying, spray drying, and other simple methods, which can achieve sustained release of a drug, targeted delivery, reduced drug use, reduced drug toxicity, and other functions [55,56,57].

### 3.3. Collagens

#### 3.3.1. Hemostatic Mechanism

Currently, the widely-recognized collagen coagulation mechanisms include: (1) collagen contains a large number of diamino dicarboxylic acid groups, has an extremely strong hydrophilicity, can absorb the wound bleeding quickly, sticks to the wound surface, forms a blood scab, and blocks the bleeding. (2) With the absorption of collagen, platelets gather in large quantities and are stimulated by collagen-related groups to release coagulation factors, thus accelerating the endogenous hemostasis process and completing the hemostasis [58].

From previous studies on the hemostatic process of marine fish collagen on skin wounds and liver wounds in rats, it was found that fish collagen could absorb and swell rapidly at the wound surface, and stopped bleeding within 30 s and formed blood scabs within 1–2 h, which was similar to the above.

#### 3.3.2. Wound Regeneration Function

Collagen is regarded as a natural biological repair material that has a similar spatial structure to the human body and which can support cell differentiation, migration, crawling, and proliferation [59]. At the same time, some scholars speculate that that its degradation products may be used by the body; therefore, we hypothesized that collagen at the wound may be decomposed and utilized in two ways (Figure 8) to promote repair [60]: (1) Collagen is recognized by macrophages as tissue fragments and devoured. After enzymatic digestion in macrophages, collagen is discharged to the wound surface [61]. (2) Collagen is decomposed by a proteolytic enzyme secreted by neutrophils to the wound surface and hydrolyzed into polypeptide. Broken down peptides are used by fibroblasts to make new collagen [62].

#### 3.3.3. Carrier Function

Studies have shown that sustained drug release and time in the body can be achieved by controlling the structure of collagen. At present, it is mainly used to react with other substances to form composite microspheres to achieve carrier functions, such as collagen–polylactic acid microspheres, collagen–hydroxyapatite microspheres, collagen–chitosan microspheres, and so on [63,64].

### 3.4. Other Functions

In addition to the above functions, the three types of marine biomaterials have a variety of applications in the medical field. For example, alginate can adsorb copper ions of tyrosinase, block the process of synthesis of melanin by tyrosinase, and whiten the skin. Chitosan can scavenge oxygen free radicals in vivo by using free amino groups, which play an antioxidant role. Collagen can increase skin elasticity, slowing the effects of ageing.

## 4. Structural Shortages of Three Polymers

With the development of medical technology and biochemistry, greater requirements have been asked of biomedical materials. In the application process of the three marine derived biological materials in the medical field, all of them have performed well, but there remain some problems to be further studied and solved for each material.

The problem with alginate is undoubtedly caused by modification. The content and proportion of M and G in the substructure of alginate determine its properties. Modification and derivatization mainly occur on the carboxyl group and hydroxyl group located on the M and G structures. It is not clear whether various modification and derivatization reactions are selective for the two structures and whether or not they can be regulated. The effects of modification on the biocompatibility and physicochemical properties of alginate have not been clarified. The degradation behavior and products of derivatives need further study.

Pure chitosan has a poor spinnability and mechanical strength and has a fast degradation rate. In a neutral pH environment, the antibacterial activity of chitosan is weak. These problems also need to be further studied and solved.

Marine collagen mainly has the following problems: (1) Poor mechanical strength and fast degradation rate in vivo. (2) There are many species to choose between and a wide range and great difference in extraction products, which need to be further clarified according to the actual needs. (3) Further studies are needed on the control of the collagen extraction process and final product. (4) The form and efficacy of the final product remain to be determined. (5) There are differences in the structure and composition of collagen and terrestrial collagen. There have been no systematic studies on the advantages and disadvantages of collagen in terms of its functionality, which needs to be confirmed before clinical applications.

## 5. Directional Modifications of Three Polymers

### 5.1. Alginate

Alginate has excellent gelling properties, hygroscopicity, hydrophilicity, etc., which make it an excellent biomedical material. However, these characteristics also have certain limitations, such as poor cell adhesion, a too strong hydrophilicity, slow degradation rate, poor mechanical properties, etc. Therefore, it needs to be modified. The modification of alginate is mainly based on the active sites of sugar units, such as –OH at C-2, C-3 and –COOH at C-6. Different active groups and molecules react with alginate to form derivatives with different functions [65]. In addition, the ratios of d-mannuronic acid (M) and l-guluronic acid (G) in alginate molecules also have a great influence on the modification reaction, which constitutes the basis of alginate modification [66].

The main modification methods shown in studies include crosslinking, grafting, acylation, carboxymethylation, and so on (Table 2). Different modification methods obtain different results. For example, the reactivity with divalent ions was increased to enhance the properties of the adhesive with sodium ions and the exchange ability of ions. Through crosslinking and recombination with other proteins, the membrane forming ability and fiber strength are enhanced [67]. Covalent crosslinking can enhance the mechanical properties and improve the stability and swelling rate of the gel. Binding with biological factors can give it cell adhesion sites and the ability to regulate cell behavior [68]. The inclusion of hydrophobic groups can reduce the hydrophilicity of alginate and effectively improve its drug carrying capacity [69]. Grafting with other polymers gives it a better mechanical and sustained-release abilities. Carrying cells and factors enhance repair functions [70,71,72].

At present, using modification technology gives many functions that are obviously better than that of single alginate products (Table 2). For example, with the use of oxidized alginate and chitosan cross-linked to obtain a hydrogel, and bovine serum albumen conjugated and cross-linked to form a microsphere water emulsion, the mechanical properties and loading sustained release performance have been greatly improved [73]. Using grafting technology, modified polyethyl methacrylate is attached to alginate to prepare high-quality sustained-release microspheres [74]. The osteogenesis-related growth factors, peptides, etc. are loaded into the alginic acid gel, so that the gel can promote bone regeneration [75]. In addition, modifications to carry different types of cells are conducive to the repair of various types of wound [76,77].

### 5.2. Chitosan

Chitosan has good biocompatibility, biodegradable, adsorption, film forming, and antimicrobial properties, as well as being non-toxic and accelerating wound healing, so it is widely used in clinical practice. However, pure chitosan’s antibacterial ability is not stable because it has poor mechanical properties and the degradation speed is too fast, and it needs to use the amino, hydroxyl, and other active sites in the molecular chain, through crosslinking, grafting, acylation, carboxy methylation, and other chemical modification methods, to generate a variety of derivatives and improve its performance (Table 3). For example, the reaction with silver ions enhanced the bacteriostatic function, while quaternary ammonium salt and hydrochloride enhanced hemostatic function with acid and salt to improve the water solubility and antioxidant capacity [8,78]. To enhance its drug carrying capacity, its molecular weight needs to increased and it must compounded with other substances. To enhance the mechanical strength, it should be compounded with organic matter, inorganic matter, or polymer materials [79,80]. For example, using the method of electrochemical precipitation, calcium ions are deposited on the chitosan scaffold, so that it has the ability for bone repair [81,82]. Chitosan modified by quaternary ammonium groups or other cations has a greatly improved antibacterial ability [83,84]. After cross-linking with sulfonate, the solubility of chitosan is significantly improved (Table 3) [85,86].

### 5.3. Collagen

Collagen is widely used as a medical biomaterial because of its low immunogenicity, good biocompatibility, and biodegradability. However, it has also some problems, such as its poor mechanical properties and fast degradation rate, which also need to be modified (Table 4). It is mainly modified by crosslinking, blending, grafting, and biomimetic mineralization (Table 4). Crosslinking with glutaraldehyde can enhance the mechanical strength and prolong the degradation time. Reaction with inorganic salts and polymer materials enhances mechanical properties. Compounding with chitosan and alginate enhances hemostatic performance. Reaction with other polymers improves drug loading capacity [87]. Hoyer et al. conducted biomimetic mineralization of salmon collagen in vitro to prepare a scaffold material that can be used for bone regeneration, and which has good mechanical properties and spatial structure, conducive to cell crawling and proliferation [88]. Nagai simultaneously cross-linked salmon collagen with 1-ethyl-3-(3-dimethylaminopropyl) carbodiimide hydrochloride (EDC) to improve its mechanical properties for periodontal bone defect treatment [28]. Another study used a chitosan and collagen preparation process composite scaffold for oral mucosal repair (Table 4) [89].

## 6. Medical Applications

Chitosan and alginate products have been used in clinical practice for many years, and a series of products have been derived according to clinical needs. There are many studies focusing on the application of marine collagen, but few clinical products have been developed. According to the query results from official websites, government websites, and technical data there are mainly the following types of products.

### 6.1. Alginate Products

Products used clinically mainly utilize its rapid gel formation and hemostatic properties. For example, gel products with hemostatic properties for various types of wound hemostasis; repair functions as filling products, used for filling sinus and cavity defects; and products with easy to gel characteristics, used for tumor embolization, arterial bleeding, organ disorders, and so on (Table 5).

### 6.2. Chitosan Products

The products using a single chitosan component mainly use its hemostatic and bacteriostatic effects. They include: (1) hemostatic products: chitosan dressings, chitosan protective film, spraying film, chitosan gel, and chitosan hemostatic powder, which are used for the hemostasis of various wound surfaces. (2) Antibacterial products: chitosan suppository, chitosan gel, chitosan anti-bacterial spray, and chitosan hydrocolloid applications, which are used for disinfection of various surfaces and internal cavities. (3) Repair products: chitosan gel, sponge, and suture line, which are used for promoting wound repair. In addition to the above products, chitosan-based artificial skin, artificial bone, microcapsule drug carriers, and other products have been widely explored and studied (Table 5).

### 6.3. Collagen Products

At present, many collagen products made from animal sources are used clinically; they mainly include repair products, such as artificial meninges, artificial skin, oral repair membranes, tissue repair membranes, and artificial bone materials [90]. The hemostatic materials (NeuSkin-F^®^, Helisorb^®^, Sheet, BioFil^®^, BioFil^®^-AB) which are produced by the Eucare Pharmaceutical company in India are the only products made with marine collagen that have been used clinically, but they also require more supporting clinical and other technological data. However, various types of biomedical materials and products based on marine derived collagen are under research (Table 5).

## 7. Potentials in Generative Medicine

As emerging biomedical materials, marine derived biomaterials are widely used in the fields of hemostasis and wound repair, and many new products based on their characteristics are being developed. Meanwhile, we also need to pay more attention to their potential problems. The questions that need to be explored and studied include: How to make biomaterials from marine derived biomaterials that have the appropriate mechanical properties, have characteristics required by clinical practice, and have degradation properties that match their functions; how to clarify the mechanism of marine derived biomaterials in medical applications, as well as the metabolites, metabolic modes, and metabolic pathways in vivo. More clinical guidance and data support are needed for developing different types of products made from marine derived biomaterials, and these will become the hotspot and direction of marine biomaterial development.

### 7.1. Soft Tissue Repair Materials

Clinical, physical, chemical, man-made, and other causes of soft tissue defects are very common, and when the defect area reaches the limit of human self-repair, we must use medical materials to assist treatment. The three marine biomaterials all have a repair function and are suitable raw materials for the preparation of soft tissue repair materials; however, some improvements must be made in terms of mechanical strength and degradability. For example, the combination of other medical polymer materials can, not only maintain the repair function, but also enhance the mechanical properties. Using 3D printing technology or other bionic technology, the soft tissue structure can be synthesized in vitro with marine biological materials as the matrix, which can greatly improve the repair effect.

### 7.2. Hard Tissue Repair Materials

Hard tissue defects caused by traffic accident, trauma, and war are on the rise year by year in the world. In clinical practice, polymer materials, inorganic materials, and metal materials are widely used, but there are certain problems. Therefore, good biological compatibility has gradually become the hotspot for application of degradable biomaterials, and marine biological materials, especially with the human body composition being similar to fish collagen. Collagen has many advantages for the preparation of hard tissue repair materials; it can be used for a variety of bionical purposes, is similar bionically to human bone as an in vitro synthesis material, and can also synthesize blood vessels, nerve biopsy materials, and effectively promote hard tissue repair.

### 7.3. Limits and Development Direction

With the improvement of medical standards, higher requirements are demanded from biomedical materials. The three marine-derived biomaterials have performed well in the application process in the medical field, but each material has some problems that need to be studied and solved.

The problem with alginate is mainly the uncertainty caused by its modification. The content and ratio of M and G in the alginate structure determine its properties. Modification and derivatization mainly occur on the carboxyl and hydroxyl groups on the M and G structure. It is not clear whether the modification and derivatization reactions of these two structures are selective and whether they can be controlled. The effect of modification on the biocompatibility and physicochemical properties of alginate is still unclear. The degradation behavior and products of derivatives need to be further studied.

Pure chitosan has a poor spinnability, low mechanical strength, and fast degradation. In a neutral pH environment, the antibacterial activity of chitosan is not strong, and these problems with chitosan need to be further studied and resolved.

Marine collagen mainly has the following problems: (1) Poor mechanical strength and rapid degradation in the body. (2) There are many kinds of fish, the differences are great, and the extracted protein is very varied, and thus further research is needed. (3) The collagen extraction process and the control of the final product need to be further studied. (4) The form and efficacy of the final product have yet to be determined. (5) The structure and composition of collagen and terrestrial collagen are different. Regarding the functional advantages and disadvantages of collagen, there is currently no systematic research, which needs to be verified before clinical application.

In addition, the development of medical materials is a long process. In order to ensure the biological safety and effectiveness of products, it is necessary to formulate standards for raw material inspection, production inspection, and finished product inspection that meet clinical needs, and it is also necessary to invest a lot of money in the production and market.

In a word, various properties and characteristics of marine biological materials have been gradually explored by humans. How to make full use of these in the medical field and develop more suitable medical products is the direction of the efforts of researchers.

## 8. Conclusions

Three polymers derived from marine organisms have excellent performance in hemostasis, repair, antibacterial action, and other aspects. There are also many products in clinical use, but the materials prepared from each macromolecule still have certain defects. The extraction process, modifications, and other aspects need to be further investigated to improve their performance and develop more products that meet clinical needs.

## Figures and Tables

**Figure 1 polymers-13-02482-f001:**
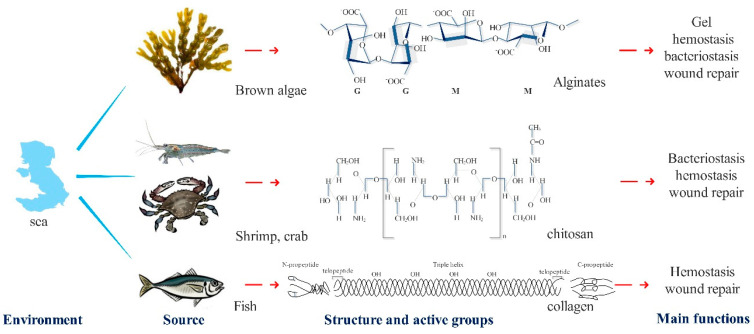
Unique structures, active groups, main functions, and source of three polymers from the sea.

**Figure 2 polymers-13-02482-f002:**
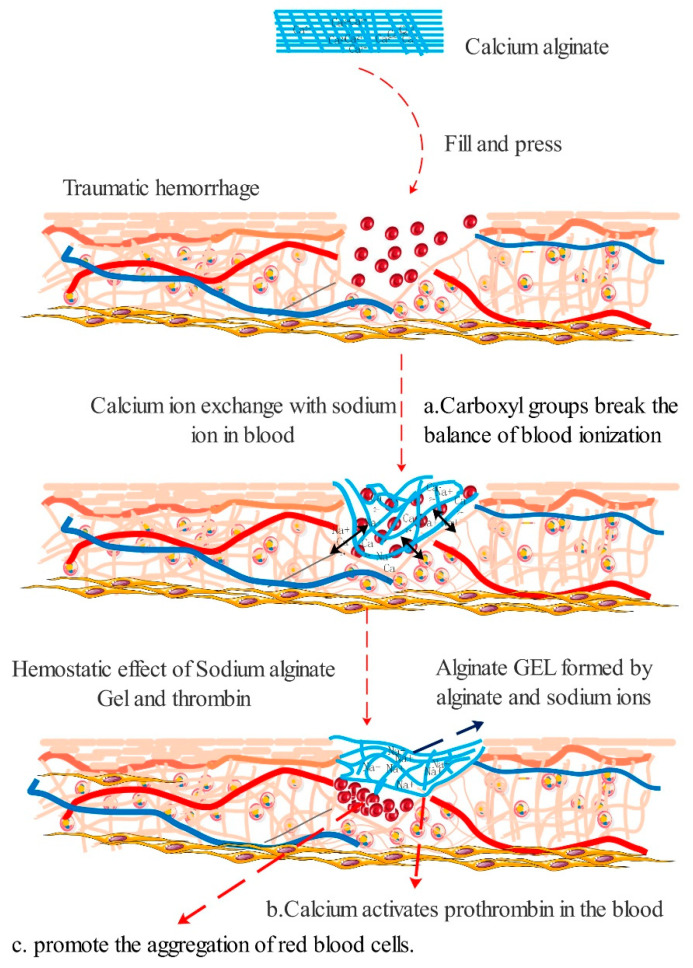
Hemostatic mechanism of calcium alginate fibers. (**a**) Carboxyl groups break the balance of blood ionization. (**b**) Calcium ions activate prothrombin (**c**) promote the aggregation of red blood cells.

**Figure 3 polymers-13-02482-f003:**
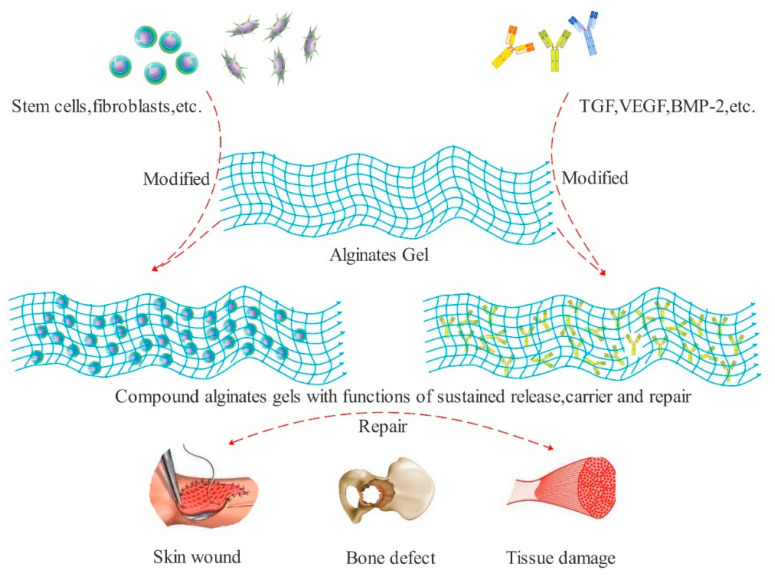
Wound regeneration mechanism of alginate gel which repairs using carry cells, factors, etc.

**Figure 4 polymers-13-02482-f004:**
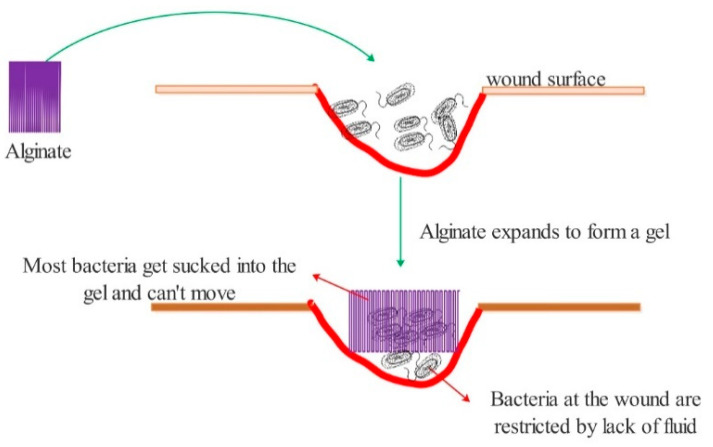
Two different antibacterial mechanisms of alginates.

**Figure 5 polymers-13-02482-f005:**
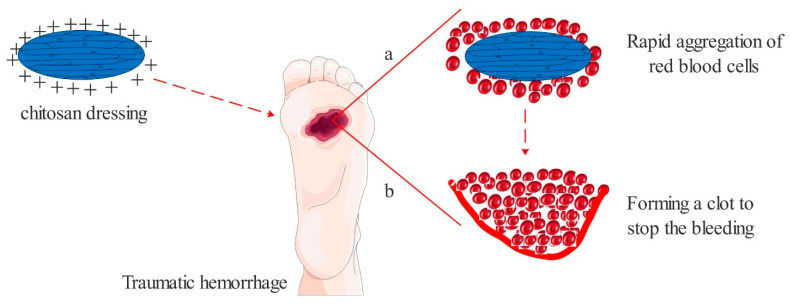
Hemostatic mechanism of chitosan. (**a**) Positive charge causes red blood cells to accumulate. (**b**) Material adheres to the wound.

**Figure 6 polymers-13-02482-f006:**
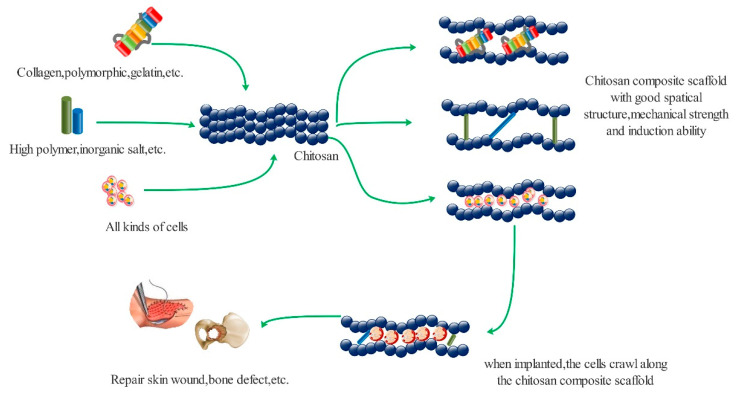
Chitosan promotes different tissue defect repair mechanisms by carrying other macromolecules.

**Figure 7 polymers-13-02482-f007:**
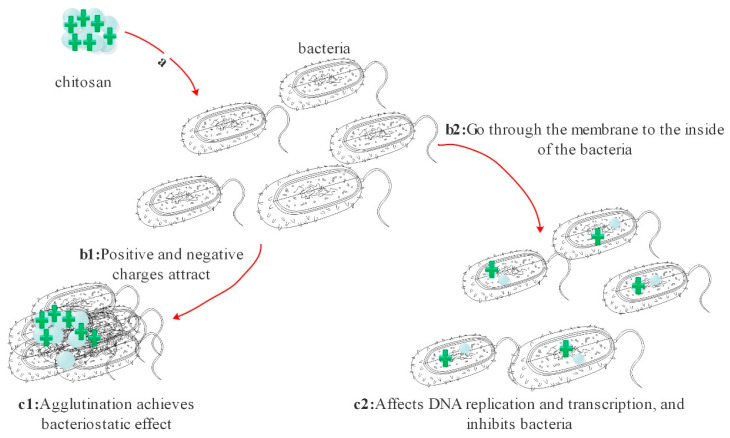
Two different antibacterial mechanisms of chitosan (**a**. Chitosan is covered on the wound. **b1**,**c1**. Cations cause bacteria to accumulate. **b2**,**c2**. effect on bacterial reproduction).

**Figure 8 polymers-13-02482-f008:**
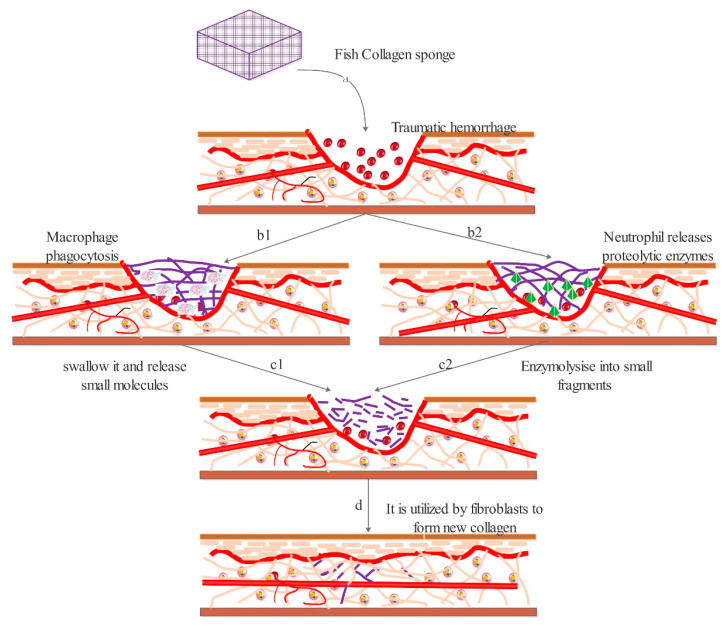
Wound regeneration mechanism of collagens (**a.** Collagen covers the wound. **b1**,**c1**. Decomposed by phagocytosis of macrophages. **b2**,**c2**. Decomposed by enzymes. **d**. Decomposed peptides participate in wound regeneration).

**Table 1 polymers-13-02482-t001:** Main active groups, functions, and sources of three polymers.

	Species	Active Groups	Maine Functions
Alginate	*Laminaria*, *Macrocystis*, *Sargassum*, etc.	d-mannuronic acid (M) l-guluronic acid (G) Free carboxyl groups Hydroxyl groups	Hemostasis [18] Bacteriostatic [17] Promote healing [19] Drug carrier [9]
Chitosan	Shrimp, crab, etc	Amino/acetyl amino Hydroxyl groups Gluco-1,4-glycoside bond	Anti-bacteria [17] Anti-inflammation [20] Anti-oxidation [21] Hemostasis [22] Wound healing [11] Drug carrier [11]
Collagen	skin, bone, scale, muscle of marine fish, like Sharks, squid, salmon, etc	Diamino-dicarboxylic groups	Wound care [20] Tissue repair [19] Tissue instead [20] Hemostasis [23] Drug delivery systems [24]

**Table 2 polymers-13-02482-t002:** Purpose, methods, and effects of directional modification of alginate.

Modification Purpose	Modification Method	Modification Effect
Enhanced gelatinization properties	Reacts with calcium ions and so on	forms a gel quickly, protects wound wetting, promotes hemostasis, carries cells and drugs [68]
Enhanced ion exchange performance	Reacts with sodium ions and so on	absorbs copper ions, lead ions and so on [70,71]
Enhances the ability of forming film and fiber	Crosslinks with ethylene oxide and blends with other proteins	Good film-forming ability, efficient moisturizing ability, and fiber ability [72,73]

**Table 3 polymers-13-02482-t003:** Modification purpose, modification method, and modification effect of chitosan.

Modification Purpose	Modification Method	Modificatin Effect
Enhanced antibacterial properties	Reaction with silver ions, deacetylation	Significantly higher [79,83]
Enhanced hemostatic property	Reaction with quaternary ammonium salt and hydrochloride or deacetylation	Greatly improved [80]
Increased antioxidant capacity	Increase water solubility and react with acids and salts	Activates antioxidant enzymes in vivo, enhances the scavenging of oxygen free radicals [81,82]
Enhanced carrier capacity	Increases molecular weight and polymerizes with other substances, such as polyethylene glycol	Carries insulin, cells, and other drugs [85]
Enhanced mechanical and inductive properties	compounded with organic compounds such as collagen, inorganic substances such as SiO_2_ and HA, and macromolecules such as polylactic acid	Improves the three-dimensional space, structure, and mechanical properties [81]

**Table 4 polymers-13-02482-t004:** Modification purpose, modification method, and modification effect of collagen.

Modification Purpose	Modification Method	Modification Effect
Enhanced hemostatic property	Compound with chitosan and other hemostatic substances	effectively improved [87]
Enhance carrier capacity	Improve the degree of polymerization, and another polymer organic compound	It can be prepared into microspheres and microcapsules [88,89]

**Table 5 polymers-13-02482-t005:** Main product forms and functions of the three polymers.

Polymers	Product Form	Functions
Alginate	Membrane, dressing, gel	Hemostasis, cavity filling, blockage [18,39]
Chitosan	Film, spray, liquid, powder, sponge, suture	Cavity filling, wound hemostasis, defect repair [2,45,49,50,51]
Collagen	Sponge, acellular matrix	Hemostasis, wound repair [26,53,66]

## Data Availability

All data included in this study are available.

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
