# Peer review of "Three Polymers from the Sea: Unique Structures, Directional Modifications, and Medical Applications"

_polymers, 2021, doi:10.3390/polym13152482_

Round 1

Reviewer 1 Report

The review article entitled ‘Three ‘Blue Polymers’ from the Sea: Unique Structures, Directional Modifications and Medical Applications in Hemostasis and Wound Repair’. The concept of the review article is interesting and suitable to publish in Polymers Journal. However, in the present form it cannot be accepted it required substantial major revision.

The major comments are as follows:

1)      Title should be modified in the precise way. What do blue polymers mean??

2)      Abstract looks very general and not informative should be rewritten. In abstract authors should mention the importance of research work briefly.

3)      A well addressed graphical scheme of study design should be inserted.  For review article it should be mandatory and don’t use the figures used in the manuscript.

4)      Introduction looks very general. In the introduction section, write the novelty of the work and the problem statement clearly. Give quantitative data of three selected polymers production and their utilization throughout the world. Add more details and substantial discussion on the slected polymers is needed.

5)      Precise research objectives and clear justification of the selection of this review topic is lacking thus major discussion is expected during revision.

6)      The figure quality is not good and clear modify each figure with high resolution.

7)      Table 2 to 4 can be combined as one table.

8)      This manuscript lacked substantial discussion of results with the recent literature authors should concentrate on this during revision.

9)       For section 5 add one table describing the commercial products of each selected polymer would be better.

11)  Techno Economic challenges of the developed composites need to be addressed by adding a new section before conclusions.

12)  What are the limitations of using these polymers for commercial application ?.

13)  Add conclusion of the study need to add with the specific output obtained from the study, it could be modified with precise outcomes with a take home message.

14)  English and grammar mistakes are present. The author should check the manuscript by native English Speaker to improve the quality of the manuscript.

Author Response

Dear reviewer,

Thank you very much for your suggestions on my article. The following are my changes to the article based on your suggestions. If you have any suggestions for my revised content, please suggest and let me know, thank you very much again.

1) Title should be modified in the precise way. What do blue polymers mean??

I had change title to “Three polymers from the sea: unique structures, directional modifications, and medical applications”.

2) Abstract looks very general and not informative should be rewritten. In abstract authors should mention the importance of research work briefly.

I have rewritten.

3) A well addressed graphical scheme of study design should be inserted.  For review article it should be mandatory and don’t use the figures used in the manuscript.

I have replaced all the pictures with my own.

4) Introduction looks very general. In the introduction section, write the novelty of the work and the problem statement clearly. Give quantitative data of three selected polymers production and their utilization throughout the world. Add more details and substantial discussion on the slected polymers is needed.

I have elaborated in the introduction

5) Precise research objectives and clear justification of the selection of this review topic is lacking thus major discussion is expected during revision.

I have added relevant content in the introduction and main text.

6) The figure quality is not good and clear modify each figure with high resolution.

I have replaced all the pictures.

7) Table 2 to 4 can be combined as one table.

The tables are too big after merging, so I didn’t merge the tables

8) This manuscript lacked substantial discussion of results with the recent literature authors should concentrate on this during revision.

I have sorted out and summarized the literature data of the past five years.

9)For section 5 add one table describing the commercial products of each selected polymer would be better.

I have added the table in this section.

11)  Techno Economic challenges of the developed composites need to be addressed by adding a new section before conclusions.

Part of the content has been added 6.3.

12)  What are the limitations of using these polymers for commercial application ?.

Related elaboration in 6.3.

13)  Add conclusion of the study need to add with the specific output obtained from the study, it could be modified with precise outcomes with a take home message.

I have added.

14)  English and grammar mistakes are present. The author should check the manuscript by native English Speaker to improve the quality of the manuscript.

I have checked.

Best wishes to you

Lei Wang

Reviewer 2 Report

The quality of the figures is not adequate and often misleading.

The quality of the review is too generic and not systematic

Bibliography is not updated

It is suggested to use a more systemic approach and  to strongly revise the paper with more recent bibliography on the interesting topic

Author Response

Dear reviewer,

Thank you very much for your suggestions on my article. The following are my changes to the article based on your suggestions. If you have any suggestions for my revised content, please suggest and let me know, thank you very much again.

1.The quality of the figures is not adequate and often misleading.

I have replaced all the pictures.

2.The quality of the review is too generic and not systematic

I have made some additions and modifications.

  1. Bibliography is not updated

I have updated

4.It is suggested to use a more systemic approach and  to strongly revise the paper with more recent bibliography on the interesting topic

I referred to the literature of the past five years and made supplements and amendments to the content

Reviewer 3 Report

Review article titled (Three ‘blue polymers’ from the sea: unique structures, directional modifications and medical applications in hemostasis and wound repair) by Wang et al. discusses 3 important blur polymers, alginate, chitosan and collaegns regarding their structures and therapeutic utility. In general this is a useful  and interesting review article. The main concern about this review is the lack of appropriate referencing. I have the following recommendations before acceptance:

1- Title: medical applications in hemostasis & wound healing: was all the 3 polymers discussed in these 2 areas? if no, please modify the title to represent all the polymers

2- Authors sometimes use (collagen) and sometimes use (collagens)

3- The over all structure needs organization: please  make a summary of the titles at the first page

4-Fig 1 resolution is poor, please enhance it to show the chemical structure well

5- I think section 5 should be before 2 & revise the whole paper to prevent redundance.

6- The references for the info in the tables are lacking

7- Many paragraphs come without references, please cite the appropriate refererences. 

Author Response

Dear reviewer,

Thank you very much for your suggestions on my article. The following are my changes to the article based on your suggestions. If you have any suggestions for my revised content, please suggest and let me know, thank you very much again.

1- Title: medical applications in hemostasis & wound healing: was all the 3 polymers discussed in these 2 areas? if no, please modify the title to represent all the polymers

I had change title to “Three polymers from the sea: unique structures, directional modifications, and medical applications”.

2- Authors sometimes use (collagen) and sometimes use (collagens)

I have changed

3- The over all structure needs organization: please make a summary of the titles at the first page

I referred to the literature of the past five years and made supplements and amendments to the content

4-Fig 1 resolution is poor, please enhance it to show the chemical structure well

I have replaced all the pictures.

5- I think section 5 should be before 2 & revise the whole paper to prevent redundance.

6- The references for the info in the tables are lacking

I have added

7- Many paragraphs come without references, please cite the appropriate refererences.

I have updated

Best wishes to you.

Lei wang

Round 2

Reviewer 1 Report

The authors have substantially revised the manuscript according to the comments.

However some information and modifications are required before its acceptance

For all figure and tables captions give all details.

Table 5 give suitable literature for the same.

Author Response

Dear reviewer,

Thank you very much for your suggestions again. According to your suggestion, I revised the article again, as follows.

  • However some information and modifications are required before its acceptance

I made further modifications to all the information, words, and sentences in the article, and reflected them in the article in a modified mode

  • For all figure and tables captions give all details.

According to your suggestion, I have described all the charts in detail in the article and the title of the chart.

  • Table 5 give suitable literature for the same.

I have given suitable literatures.

Thank you very much.

Best wishes to you.

Lei Wang

Reviewer 2 Report

The paper has been sufficiently updated according to my suggestions

Author Response

Dear reviewer,

Thank you for your valuable comments on my article again.

Best wishes to you.

Lei Wang

Reviewer 3 Report

thanks for the authors for providing a good revised version

Author Response

(The authors gave the same response as above.)
